# HiCL: Hierarchical Contrastive Learning of Unsupervised Sentence Embeddings

**Zhuofeng Wu**
University of Michigan
Ann Arbor
zhuofeng@umich.edu

**Chaowei Xiao**
University of Wisconsin
Madison
cxiao34@wisc.edu

**VG Vinod Vydiswaran**
University of Michigan
Ann Arbor
vgvinodv@umich.edu

## Abstract

In this paper, we propose a hierarchical contrastive learning framework, HiCL, which considers local segment-level and global sequence-level relationships to improve training efficiency and effectiveness. Traditional methods typically encode a sequence in its entirety for contrast with others, often neglecting local representation learning, leading to challenges in generalizing to shorter texts. Conversely, HiCL improves its effectiveness by dividing the sequence into several segments and employing both local and global contrastive learning to model segment-level and sequence-level relationships. Further, considering the quadratic time complexity of transformers over input tokens, HiCL boosts training efficiency by first encoding short segments and then aggregating them to obtain the sequence representation. Extensive experiments show that HiCL enhances the prior top-performing SNCSE model across seven extensively evaluated STS tasks, with an average increase of +0.2% observed on BERT$_{large}$ and +0.44% on RoBERTa$_{large}$. [1]

## 1 Introduction

Current machine learning systems benefit greatly from large amounts of labeled data. However, obtaining such labeled data is expensive through annotation in supervised learning. To address this issue, self-supervised learning, where supervisory labels are defined from the data itself, has been proposed. Among them, contrastive learning (Chen et al., 2020a,b,c, 2021; He et al., 2020; Grill et al., 2020; Chen and He, 2021) has become one of the most popular self-supervised learning methods due to its impressive performance across various domains. The training target of contrastive learning is to learn a representation of the data that maximizes the similarity between positive examples and minimizes the similarity between negative examples.

To achieve better performance, existing methods mainly focus on designing better positive examples (Hendrycks et al., 2020; Fang et al., 2020; Wu et al., 2020; Giorgi et al., 2021; Gao et al., 2021b) or investigating the role of the negative examples (Robinson et al., 2020; Zhou et al., 2022; Wang et al., 2022).

Despite the success, existing methods augment data at the level of the full sequence (Gao et al., 2021b; Wang et al., 2022). Such methods require calculating the entire sequence representation, leading to a high computational cost. Additionally, it also makes the task of distinguishing positive examples from negative ones too easy, which doesn't lead to learning meaningful representations. Similarly, methods like CLEAR (Wu et al., 2020) demonstrated that pre-training with sequence-level naïve augmentation can cause the model to converge too quickly, resulting in poor generalization.

In contrast, Zhang et al. (2019) considered modeling in-sequence (or local) relationships for language understanding. They divide the sequence into smaller segments to learn intrinsic and underlying relationships within the sequence. Since this method is effective in modeling long sequences by not truncating the input and avoiding loss of information, it achieves promising results. Given this success, a natural question arises: is it possible to design an effective and efficient contrastive learning framework by considering the local segment-level and global sequence-level relationships?

To answer the question, in this paper, we propose a hierarchical contrastive learning framework, HiCL, which not only considers global relationships but also values local relationships, as illustrated in Figure 1. Specifically, given a sequence (i.e., sentence), HiCL first divides it into smaller segments and encodes each segment to calculate local segment representation respectively. It then aggregates the local segment representations belonging to the same sequence to get the global

---

[1] Our code will be released at https://github.com/CSerxy/HiCL.

sequence representation. Having obtained local and global representations, HiCL deploys a hierarchical contrastive learning strategy involving both segment-level and sequence-level contrastive learning to derive an enhanced representation. For local contrastive learning, each segment is fed into the model twice to form the positive pair, with segments from differing sequences serving as the negative examples. For global contrastive learning, HiCL aligns with mainstream baselines to construct positive/negative pairs.

We have carried out extensive experiments on seven STS tasks using well-representative models BERT and RoBERTa as our backbones. We assess the method's generalization capability against three baselines: SimCSE, ESimCSE, and SNCSE. As a result, we improve the current state-of-the-art model SNCSE over seven STS tasks and achieve new state-of-the-art results. Multiple initializations and varied training corpora confirmed the robustness of our HiCL method.

**Our contributions** are summarized below:

- To the best of our knowledge, we are the first to explore the relationship between local and global representation for contrastive learning in NLP.
- We theoretically demonstrate that the encoding efficiency of our proposed method is much faster than prior contrastive training paradigms.
- We empirically verify that the proposed training paradigm enhances the performance of current state-of-the-art methods for sentence embeddings on seven STS tasks.

## 2 Preliminaries: Contrastive Learning

In this paper, we primarily follow SimCLR's framework (Chen et al., 2020a) as our basic contrastive framework and describe it below. The general training objective of contrastive learning (Oord et al., 2018) is to distinguish similar pairs from dissimilar pairs, where similar pairs are constructed using pre-defined data augmentation techniques and dissimilar pairs are other examples in the same batch. Specifically, for an arbitrary example $x_i$ in a batch $B$, the InfoNCE loss $\mathcal{L}_g$ brings the representation $h_i$ closer to positive instance representation $h_i^+$ and away from negative ones $h_{j \in B \setminus i}^-$. If $h_i, h_i^+, h_{j \in B \setminus i}^-$ are the representation vectors from the encoder, $\tau$ is a temperature scale factor (set to 0.05 following SimCSE), and $sim(u, v) = u^T v / \|u\| \|v\|$ denotes the cosine similarity, $\mathcal{L}_g$ is computed as:

$$\mathcal{L}_g = -\log \frac{e^{sim(h_i, h_i^+)/\tau}}{e^{sim(h_i, h_i^+)/\tau} + \sum_{j \in B \setminus i} e^{sim(h_i, h_j)/\tau}},$$

(1)

Benefiting from human-defined data augmentations, it can generate numerous positive and negative examples for training without the need for explicit supervision, which is arguably the key reason why self-supervised learning can be effective.

**Positive instance** Designing effective data augmentations to generate positive examples is a key challenge in contrastive learning. Various methods such as back-translation, span sampling, word deletion, reordering, and synonym substitution have been explored for language understanding tasks in prior works such as CERT (Fang et al., 2020), DeCLUTR (Giorgi et al., 2021), and CLEAR (Wu et al., 2020). Different from previous approaches that augment data at the discrete text level, SimCSE (Gao et al., 2021b) first applied dropout (Srivastava et al., 2014) twice to obtain two intermediate representations for a positive pair. Specifically, given a Transformers model, $E_\theta$ (parameterized by $\theta$), (Vaswani et al., 2017) and a training instance $x_i$, $h_i = E_{\theta,p}(x_i)$ and $h_i^+ = E_{\theta,p^+}(x_i)$ are the positive pair that can be used in Eq. 1, where $p$ and $p^+$ are different dropout masks. This method has been shown to significantly improve sentence embedding performance on seven STS tasks, making it a standard comparison method in this field.

**Negative instance** Negative instance selection is another important aspect of contrastive learning. SimCLR simply uses all other examples in the same batch as negatives. However, in DCLR (Zhou et al., 2022), around half of in-batch negatives were similar to SimCSE's training corpus (with a cosine similarity above 0.7). To address this issue, SNCSE (Wang et al., 2022) introduced the use of negated sentences as "soft" negative samples (e.g., by adding *not* to the original sentence). Additionally, instead of using the [CLS] token's vector representation, SNCSE incorporates a manually designed prompt: "The sentence of $x_i$ means [MASK]" and takes the [MASK] token's vector to represent the full sentence $x_i$. This approach has been shown to improve performance compared to using the [CLS] token's vector. In this paper, we compare against SNCSE as another key baseline, not only because it is the current state-of-the-art

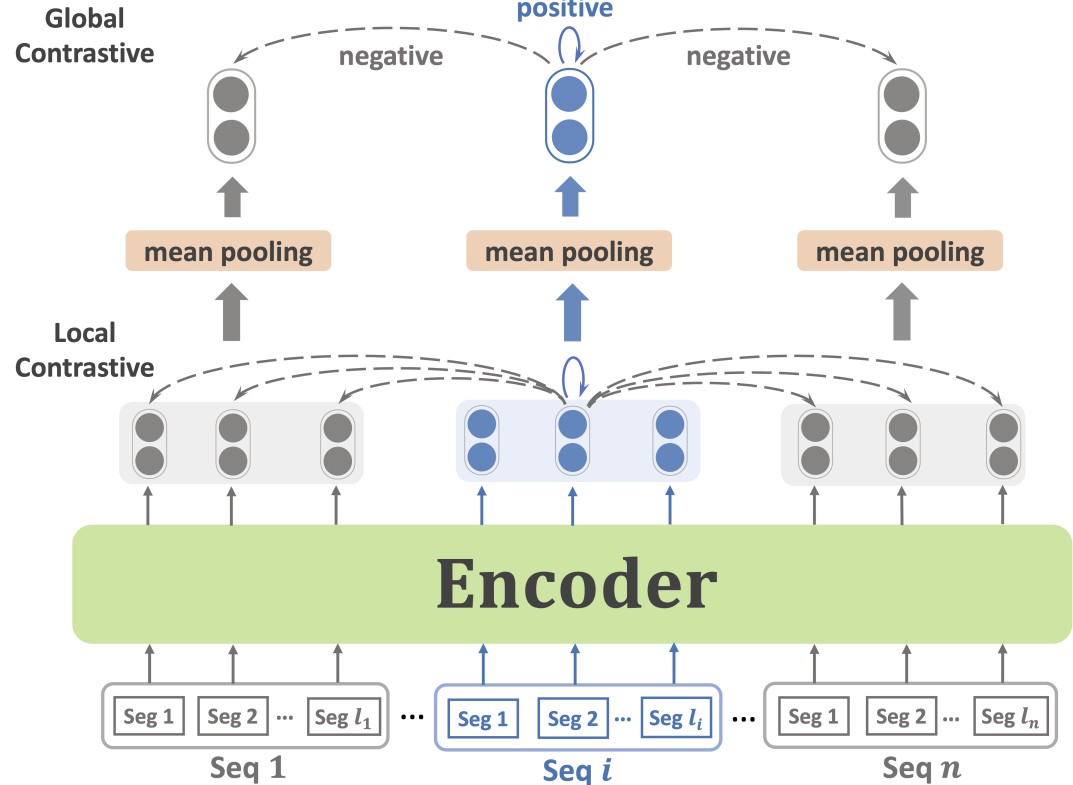

Figure 1: The overview of the HiCL framework with local contrastive and global contrastive objective.

on evaluation tasks, but also because it effectively combines contrastive learning with techniques like prompt tuning (Gao et al., 2021a; Wu et al., 2022c).

**Momentum Contrast** The momentum contrast framework differs from SimCLR by expanding the negative pool through the inclusion of recent instances, effectively increasing the batch size without causing out-of-memory issues. ESimCSE (Wu et al., 2022b) proposes a repetition operation to generate positive instances and utilizes momentum contrast to update the model. We include it as a baseline for comparison to assess the ability of our model to adapt to momentum contrast.

## 3 Hierarchical Contrastive Learning

### 3.1 Overview

Figure 1 shows an overview of HiCL. Our primary goal is to incorporate additional underlying (local) information in traditional, unsupervised text contrastive learning. Two objectives are combined to achieve this goal.

Given a set of sequences $\{\mathsf{seq}_1, \mathsf{seq}_2, \dots, \mathsf{seq}_n\}$ in a batch $B$, we slice each $\mathsf{seq}_i$ into segments $\{\mathsf{seg}_{i,1}, \mathsf{seg}_{i,2}, \dots, \mathsf{seg}_{i,l_i}\}$ of slicing length $L$, where $n$ is the batch size, and $l_i = 1 + \lfloor (|\mathsf{seq}_i| - 1)/L \rfloor$ is the number of

segments that can be sliced in $\mathsf{seq}_i$. The slicing is performed using a queue rule: every consecutive $L$ tokens (with no overlap) group as one segment, and the remaining tokens with length no greater than $L$ form a separate segment. In other words, $|\mathsf{seg}_{i,j}| = L, \forall j \in [1, l_i); |\mathsf{seg}_{i,l_i}| \in [1, L]$; and $\mathsf{seq}_i = \mathsf{concat}[\mathsf{seg}_{i,1}, \dots, \mathsf{seg}_{i,l_i}]$.

Unlike traditional contrastive learning, which encodes the input sequence $\mathsf{seq}_i$ directly, we encode each sub-sequence $\mathsf{seg}_{i,j}$ using the same encoder and obtain its representation: $h_{i,j} = E_\theta(\mathsf{seg}_{i,j})$, where $E_\theta$ is a Transformer (Vaswani et al., 2017), parameterized by $\theta$. We aggregate the $h_{i,j}$ representations to obtain the whole sequence representation $h_i$ by weighted average pooling, where the weight of each segment $\mathsf{seg}_{i,j}$ is proportional to its length $|\mathsf{seg}_{i,j}|$: $h_i = \sum_j h_{i,j} \times w_{i,j}$, where $w_{i,j} = \frac{|\mathsf{seg}_{i,j}|}{\sum_k |\mathsf{seg}_{i,k}|}$. In Section 5.1, we explore other pooling methods, such as unweighted average pooling, and find that weighted pooling is the most effective. According to Table 2, most (99.696%) input instances can be divided into three or fewer segments. Therefore, we do not add an extra transformer layer to get the sequence representation from these segments, as they are relatively short.

To use HiCL with SNCSE, we slice the input se-

quence in the same way, but add the prompt to each segment instead of the entire sequence. We also apply the same method to the negated sentences.

## 3.2 Training Objectives

**Local contrastive**   Previous studies have highlighted the benefits of local contrastive learning for unsupervised models (Wu et al., 2020; Giorgi et al., 2021). By enabling the model to focus on short sentences, local contrastive learning allows the model to better match the sentence length distribution, as longer sentences are less common. Building on the work of Gao et al. (2021b), we use dropout as a minimum data augmentation technique. We feed each segment $\text{seg}_{i,j}$ twice to the encoder using different dropout masks $p$ and $p^+$. This results in positive pairs $h_{i,j} = E_{\theta,p}(\text{seg}_{i,j})$ and $h_{i,j}^+ = E_{\theta,p^+}(\text{seg}_{i,j})$ for loss computation. As mentioned in Section 1, defining negatives for segments can be challenging. Using segments from the same sequence as negatives carries the risk of introducing correlations, but treating them as positive pairs is not ideal either. We chose not to use segments from the same sequence as either positive or negative pairs and we will show that this approach is better than the other alternatives in Section 5.2. Hence, for segment $\text{seg}_{i,j}$, we only consider as negatives, segments from other sequences $\{\text{seg}_{k,*}, k \in \{B \setminus i\}\}$. The local contrastive $\mathcal{L}_l$ is formalized as:

$$\mathcal{L}_l = -\log \frac{e^{sim(h_{i,j}, h_{i,j}^+)/\tau}}{e^{sim(h_{i,j}, h_{i,j}^+)/\tau} + \sum_{k \neq i} e^{sim(h_{i,j}, h_{k,*})/\tau}}$$
(2)

**Global contrastive**   The global contrastive objective is the same as that used by most baselines, which tries to pull a sequence's representation $h_i$ closer to its positive $h_i^+$ while keeping it away from in-batch negatives $h_{j \in B \setminus i}^-$, as defined by the global contrastive loss $\mathcal{L}_g$ in Eq. 1.

**Overall objective**   The overall objective is a combination of local and global contrastive loss, $\mathcal{L} = \alpha \mathcal{L}_l + (1 - \alpha)\mathcal{L}_g$, where weight $\alpha \in \{0.01, 0.05, 0.15\}$ is tuned for backbone models. Our adoption of a hybrid loss, with a lower weighting assigned to the local contrastive objective, is motivated by the potential influence of the hard truncation process applied to the sequences. This process can result in information loss and atypical sentence beginnings that may undermine the effectiveness of the local contrastive loss. Meanwhile, a

standalone global contrastive loss is equally inadequate, as it omits local observation. We conduct an analysis in Section 5.4 to discuss the intricate relationship between two objectives.

## 3.3 Encoding Time Complexity Analysis

According to our slicing rule, all front segments $\text{seg}_{i,j<l_i}$ in sequence $\text{seq}_i$ have length $L$ and the last segment $\text{seg}_{i,l_i}$ has length $|\text{seg}_{i,l_i}| \in [1, L]$. Hence, the encoding time complexity for HiCL is $O(L^2 \times (l_i - 1) + |\text{seg}_{l_i}|^2)$, while the conventional methods take:

$$O(|\text{seq}_i|^2) = O((L \times (l_i - 1) + |\text{seg}_{l_i}|)^2)$$
$$> (l_i - 1) \times O(L^2(l_i - 1) + |\text{seg}_{l_i}|^2)$$

which is $(l_i - 1)$ times more than that for HiCL. The longer the training corpus, the higher the benefit using HiCL. This suggests that our approach has a variety of use cases, particularly in pre-training, due to its efficient encoding process.

The practical training time is detailed in Appendix A.1. In short, we are faster than baselines when maintaining the same sequence truncation size 512. For example, SimCSE-RoBERTa$_{\text{large}}$ takes 354.5 minutes for training, while our method only costs 152 minutes.

## 4 Experiment

### 4.1 Experimental Setup

**Evaluation tasks**   Following prior works (Gao et al., 2021b; Wang et al., 2022), we mainly evaluate on seven standard semantic textual similarity datasets: STS12–16 (Agirre et al., 2012, 2013, 2014, 2015, 2016), STS Benchmark (Cer et al., 2017), and SICK-Relatedness (Marelli et al., 2014). When evaluating using the SentEval tookit[2], we adopt SimCSE's evaluation setting without any additional regressor and use Spearman's correlation as the evaluation matrix. Each method calculates the similarity, ranging from 0 to 5, between sentence pairs in the datasets. In our ablation study, we extend our evaluation to encompass two lengthy datasets (i.e., Yelp (Zhang et al., 2015) and IMDB (Maas et al., 2011)) and seven transfer tasks (Conneau and Kiela, 2018). Due to space consideration, we have provided detailed results for the seven transfer learning tasks in Appendix A.2.

---

[2]https://github.com/facebookresearch/SentEval

Table 1: Main results of various contrastive learning methods on seven semantic textual similarity (STS) datasets. Each method is evaluated on full test sets by Spearman's correlation, "all" setting. **Bold** marks the best result among all competing methods under the same backbone model.

| Method | STS12 | STS13 | STS14 | STS15 | STS16 | STS-B | SICK-R | Avg. |
|---|---|---|---|---|---|---|---|---|
| SimCSE-BERT$_{base}$ | 66.92 | 78.44 | 71.15 | 79.50 | 77.71 | 75.50 | 68.76 | 74.00 |
| + HiCL | 69.04 | 80.68 | 72.71 | 80.39 | 78.68 | 76.96 | 70.35 | 75.54↑ |
| ESimCSE-BERT$_{base}$ | **70.95** | 82.76 | 76.52 | 83.18 | 80.03 | 80.15 | 71.07 | 77.81 |
| + HiCL | 70.05 | 83.18 | 76.51 | 82.97 | **80.31** | 80.02 | 72.53 | 77.94↑ |
| SNCSE-BERT$_{base}$ | 70.15 | 84.36 | 76.86 | **83.21** | 80.16 | **81.09** | **75.04** | **78.70** |
| + HiCL | 70.57 | **84.51** | **76.92** | 82.97 | 79.91 | 80.68 | 74.61 | 78.60 |
| SimCSE-BERT$_{large}$ | 70.17 | 83.27 | 72.82 | 82.53 | 77.05 | 78.25 | 66.01 | 75.73 |
| + HiCL | 70.99 | 85.29 | 75.62 | 83.56 | 79.40 | 79.91 | 74.13 | 78.41↑ |
| ESimCSE-BERT$_{large}$ | 71.43 | 83.91 | 75.88 | 83.84 | 78.86 | 79.74 | 73.59 | 78.18 |
| + HiCL | **72.40** | 85.42 | 77.29 | 84.59 | 79.78 | 80.72 | 74.28 | 79.21↑ |
| SNCSE-BERT$_{large}$ | 72.03 | **86.80** | 78.48 | 85.27 | 80.65 | **82.20** | 74.40 | 79.98 |
| + HiCL | **72.40** | 86.78 | **78.50** | **85.52** | **80.85** | 82.09 | **75.13** | **80.18**↑ |
| SimCSE-RoBERTa$_{base}$ | 69.47 | 82.12 | 73.96 | 82.48 | 81.11 | 81.06 | 69.45 | 77.09 |
| + HiCL | 69.36 | 81.77 | 73.75 | 82.58 | 81.03 | 81.06 | 69.58 | 77.02 |
| ESimCSE-RoBERTa$_{base}$ | 69.03 | 80.73 | 73.35 | 81.26 | 81.25 | 80.26 | 68.11 | 76.28 |
| + HiCL | 69.98 | 81.31 | 74.39 | 82.87 | 81.43 | 80.74 | 69.09 | 77.12↑ |
| SNCSE-RoBERTa$_{base}$ | 70.50 | **83.70** | **76.55** | **84.09** | **81.89** | 81.73 | **74.11** | **78.94** |
| + HiCL | **71.01** | **83.70** | 76.08 | 84.06 | **81.89** | 82.07 | 73.54 | 78.91 |
| SimCSE-RoBERTa$_{large}$ | 71.41 | 83.90 | 76.50 | 85.07 | 82.10 | 82.75 | 71.17 | 78.99 |
| + HiCL | 72.36 | 84.08 | 76.35 | 85.07 | 82.49 | 82.95 | 71.40 | 79.24↑ |
| ESimCSE-RoBERTa$_{large}$ | 70.92 | 83.25 | 75.62 | 81.91 | 80.16 | 81.73 | 72.03 | 77.95 |
| + HiCL | 73.06 | 84.44 | 76.56 | 84.59 | 81.42 | 83.61 | 71.49 | 79.31↑ |
| SNCSE-RoBERTa$_{large}$ | 73.65 | **86.45** | 79.23 | 86.60 | 82.45 | 83.95 | 77.12 | 81.35 |
| + HiCL | **73.74** | 86.19 | **79.76** | **86.70** | **83.20** | **84.52** | **78.45** | **81.79**↑ |

**Implementing competing methods** We re-train three previous state-of-the-art unsupervised sentence embedding methods on STS tasks – SimCSE (Gao et al., 2021b), ESimCSE (Wu et al., 2022b), and SNCSE (Wang et al., 2022) – with our novel training paradigm, HiCL. We employ the official implementations of these models and adhere to the unsupervised setting utilized by Sim-CSE. This involves training the models on a dataset comprising 1 million English Wikipedia sentences for a single epoch. More detailed information can be found in Appendix A.3. We also carefully tune the weight $\alpha$ of local contrastive along with the learning rate in Appendix A.4.

### 4.2 Main Results

Table 1 presents the results of adding HiCL to various baselines on the seven STS tasks. We observe that (i) HiCL consistently improves the performance of baselines in the large model setting. Specifically, HiCL improves the average scores by +0.25%, +1.36%, and +0.44% on the RoBERTa$_{large}$ variants of SimCSE, ESimCSE, and SNCSE baselines, respectively. (ii) HiCL with SNCSE-RoBERTa$_{large}$ as the backbone has achieved a new state-of-the-art score of 81.79. It is worth mentioning that SNCSE originally reported a score of 81.77 based on full precision (fp32), but we achieved a better result with half precision (fp16). (iii) HiCL enhances the performance across nine backbone models, yet marginally underperforms ($\leq 0.1$) on three models employing the -base architecture.

All prior studies in the field have followed a common practice of employing the same random seed for a single run to ensure fair comparisons. We have rigorously adhered to this convention when presenting our findings in Table 1. However, to further assess the robustness of our method, we have extended our investigation by conducting multiple runs with varying random seeds. As shown in Table 15, we generally observe consistency between the multi-run results and the one-run results. The nine backbone models on which HiCL demonstrated superior performance under the default random seed continue to be outperformed by HiCL.

| Input Length | [0, 32] | (32, 64] | (64, 96] |
|---|---|---|---|
| Proportion | 76.031% | 21.869% | 1.796% |
| Input Length | (96, 128] | (128, 256] | (256, 512] |
| Proportion | 0.225% | 0.075% | 0.004% |

Table 2: The input length distribution of training corpus. Special tokens [CLS] and [SEP] counted.

# 5 Intrinsic Study and Discussion

## 5.1 Local Information Aggregation

Forming the representation for the entire sequence from segment representations is crucial to our task. We experiment with both weighted average pooling (weighted by the number of tokens in each segment) and unweighted average pooling. Since most sequences are divided into three or fewer segments (as shown in Table 2), we did not include an additional layer (either DNN or Transformer) to model relationships with $\leq 3$ inputs. Therefore, we opted to not consider aggregating through a deep neural layer, even though this approach might work in scenarios where the training sequences are longer. Results in Table 3 indicate that under three different backbones, weighted pooling is a better strategy for extracting the global representation from local segments.

| Backbone | Pooling | SICK-R |
|---|---|---|
| SimCSE-RoBERTa$_{large}$ | unweighted | 73.91 |
| | weighted | **75.24** |
| ESimCSE-RoBERTa$_{large}$ | unweighted | 73.13 |
| | weighted | **74.37** |
| SNCSE-RoBERTa$_{large}$ | unweighted | 79.97 |
| | weighted | **80.86** |

Table 3: Development set results of SICK-R (Spearman's correlation) for different pooling matrices.

## 5.2 Relationships between Segments from Same Sequence

When optimizing the local contrastive objective, an alternative approach is to follow the traditional training paradigm, which treats all other segments as negatives. However, since different parts of a sequence might describe the same thing, segments from the same sequence could be more similar compared to a random segment. As a result, establishing the relationship between segments from the same sequence presents a challenge. We explore

with three different versions: 1) considering them as positive pairs, 2) treating them as negative pairs, and 3) categorizing them as neither positive nor negative pairs. The results in Table 4 of SimCSE and SNCSE indicate that the optimal approach is to treat them as neither positive nor negative - a conclusion that aligns with our expectations.

An outlier to this trend is observed in the results for ESimCSE. We postulate that this anomaly arises due to ESimCSE's use of token duplication to create positive examples, which increases the similarity of segments within the same sequence. Consequently, this outcome is not entirely unexpected. Given that ESimCSE's case is special, we believe that, in general, it is most appropriate to label them as neither positive nor negative.

| Backbone | Positive | Negative | Neither |
|---|---|---|---|
| SimCSE-RoBERTa$_{large}$ | 73.92 | 74.34 | **75.24** |
| ESimCSE-RoBERTa$_{large}$ | **76.96** | 74.35 | 74.37 |
| SNCSE-RoBERTa$_{large}$ | 77.33 | 79.33 | **80.86** |

Table 4: Development set results of SICK-R (Spearman's correlation) for processing relationship of segments from same sequence.

## 5.3 Optimal Slicing Length

To verify the impact of slicing length on performance, we vary the slicing length from 16 to 40 for HiCL in SimCSE-RoBERTa$_{large}$ and SNCSE-RoBERTa$_{large}$ settings (not counting prompts). We were unable to process longer lengths due to memory limitations. From the results in Figure 2, we find that using short slicing length can negatively impact model performance. As we showed in Section 3.3, the longer the slicing length, the slower it is to encode the whole input sequence, because longer slicing lengths mean fewer segments. Therefore, we recommend setting the default slicing length as 32, as it provides a good balance between performance and efficiency.

We acknowledge that using a non-fixed slicing strategy, such as truncation by punctuation, could potentially enhance performance. Nevertheless, in pursuit of maximizing encoding efficiency, we have opted for a fixed-length truncation approach, leaving the investigation of alternative strategies to future work.

## 5.4 How Hierarchical Training Helps?

One might argue that our proposed method could benefit from the truncated portions of the training

Table 5: Performance on seven STS tasks for methods trained on wiki-103 and 1 million English Wikipedia sentences. Each method is evaluated on full test sets by Spearman's correlation, "all" setting.

| Method | STS12 | STS13 | STS14 | STS15 | STS16 | STS-B | SICK-R | Avg. |
|---|---|---|---|---|---|---|---|---|
| SimCSE-RoBERTa$_{large}$ | 71.72 | 84.33 | 76.45 | 84.16 | 80.36 | 81.90 | **72.27** | 78.74 |
| + HiCL | **72.45** | **84.69** | **76.98** | **84.79** | **81.64** | **82.43** | 71.54 | **79.22** |

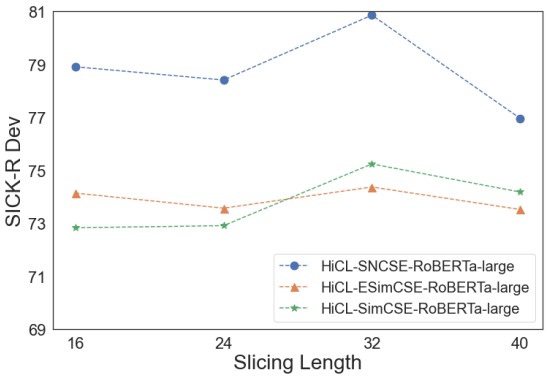

Figure 2: Comparison between different slicing lengths over three backbone models by Spearman's correlation.

corpus - the part exceeding the length limitations of baselines and thus, unprocessable by them. To address this concern, we reconstruct the training corpus in such a way that any sequence longer than the optimal slicing length (32) is divided and stored as several independent sequences, each of which fits within the length limit. This allows the baseline models to retain the same information as HiCL. The aforementioned models are indeed using a local-only loss, given they implement contrastive loss on the segmented data. Table 6 shows that HiCL continues to exceed the performance of single segment-level contrastive learning models, indicating its superior performance is not solely reliant on the reduced data. The lower performance exhibited by these models effectively emphasizes the significance of incorporating a hybrid loss.

| Method | vanilla | + HiCL | *full-* |
|---|---|---|---|
| SimCSE-RoBERTa$_{large}$ | 78.99 | **79.24** | 78.45 |
| SNCSE-RoBERTa$_{large}$ | 81.35 | **81.79** | 80.41 |

Table 6: Average performance of baselines trained with full segments (abbreviated as *full-*) on seven STS tasks.

The intriguing insight from above observations is that the omitted data does not improve the performance of the baseline models; in fact, it hinders performance. We hypothesize that this is due to a hard cut by length resulting in some segments beginning with unusual tokens, making it more difficult for the encoder to accurately represent their meaning.

We further verify this hypothesis by doing an ablation study on HiCL with various values of $\alpha$ in the overall objective. Recall that our training objective is a combination of local contrastive and global contrastive loss. HiCL model with $\alpha = 0$ is identical to the baselines, except that it incorporates the omitted information. As shown in Table 7, training with just global contrastive loss with complete information yields poorer results. Similarly, when $\alpha = 1$, the HiCL model focuses solely on local contrastive loss, but also performs poorly, which indicates that global contrastive loss is an essential component in learning sentence representation.

| $\alpha$ | 0.0 | 0.01 | 0.05 | 0.15 |
|---|---|---|---|---|
| **SICK-R** | 75.10 | 75.19 | **75.24** | 74.07 |
| $\alpha$ | 0.3 | 0.5 | 0.95 | 1.0 |
| **SICK-R** | 73.44 | 74.68 | 74.87 | 74.39 |

Table 7: Ablation study over $\alpha$ on SimCSE-RoBERTa-large at SICK-R dev set (Spearman's correlation).

It's crucial to clarify that our approach isn't just a variant of the baseline with a larger batch. As Table 2 indicates, in 99.9% of instances, the training data can be divided into four or fewer segments. Comparing SimCSE-BERT$_{base}$ with a batch size quadruple that of our method (as shown in Table 10), it's evident that SimCSE at a batch size of 256 trails our model's performance with a batch size of 64 (74.00 vs. 76.35). Simply amplifying the batch size for baselines also leads to computational issues. For example, SimCSE encounters an "Out of Memory" error with a batch size of 1024, a problem our model, with a batch size of 256, avoids. Therefore, our approach is distinct and offers benefits beyond merely adjusting batch sizes.

## 5.5 HiCL on Longer Training Corpus

To further verify the effectiveness of HiCL, we add a longer corpus, WikiText-103, along with the original 1 million training data. WikiText-103 is a dataset that contains 103 million tokens from 28,475 articles. We adopt a smaller batch size of 64 to avoid out-of-memory issue. Other training

details followed the instructions in Section 4.1. As shown in Table 5, HiCL shows more improvement (+0.48%) compared to the version only trained on short corpus (+0.25%). This indicates that HiCL is more suitable for pre-training scenarios, particularly when the training corpus is relatively long.

## 5.6 HiCL on Longer Test Datasets

The datasets that were widely evaluated, such as STS12-16, STS-B, and SICK-R, primarily consist of short examples. However, given our interest in understanding how HiCL performs on longer and more complex tasks, we further conduct evaluations on the Yelp (Zhang et al., 2015) and IMDB (Maas et al., 2011) datasets. Table 8 provides an overview of these two datasets.

Specifically, we test with SimCSE-BERT$_{base}$ backbone model and follow the evaluation settings outlined in SimCSE, refraining from any further fine-tuning on Yelp and IMDB. The results are compelling, with our proposed method consistently outperforming SimCSE, achieving a performance gain of +1.97% on Yelp and +2.27% on IMDB.

| Datasets | #Test | Length | SimCSE | +HiCL |
|---|---|---|---|---|
| Yelp | 38K | 132.6 | 70.99 | **72.96** |
| IMDB | 25K | 228.5 | 60.93 | **63.20** |

Table 8: Statistics and evaluation results on Yelp and IMDB datasets.

## 5.7 A Variant of HiCL

As we discussed in Section 5.2, the best approach to treat relationships between in-sequence segments is to consider them as neither positive nor negative. However, this would result in losing information about them belonging to the same sequence. To overcome this, we consider modeling the relationship between sequences and segments. Since each segment originates from a sequence, they inherently contain an entailment relationship – the sequence entails the segment. We refer to this variant as HiCLv2. Additional details are provided in Appendix A.6.

As shown in Table 9, explicitly modeling this sequence-segment relationship does not help the model. We think that it is probably because this objective forces the representation of each sequence to be closer to segments from the same sequence. When the representation of each sequence is a weighted average pooling of segments, it pulls segments from the same sequence closer, which is

another way of regarding them as positive. As seen in the results in Section 5.2, treating segments from the same sequence as positive would negatively impact the performance of SimCSE and SNCSE backbones. Thus, it is not surprising that HiCLv2 failed to show as much improvement as HiCL.

| | | + HiCL | + HiCLv2 |
|---|---|---|---|
| | SimCSE-BERT$_{large}$ | | |
| Avg | 75.73 | **78.41** | 76.30 |
| | SNCSE-BERT$_{large}$ | | |
| Avg | 79.98 | **80.18** | 80.00 |
| | SimCSE-RoBERTa$_{large}$ | | |
| Avg | 78.99 | **79.24** | 79.08 |
| | SNCSE-RoBERTa$_{large}$ | | |
| Avg | 81.35 | **81.79** | 81.49 |

Table 9: Average performance of HiCLv2 over seven STS datasets.

## 5.8 Baseline Reproduction

One might wonder why there are notable discrepancies between our reproduced baselines and the numbers reported in the original papers. For instance, our SimCSE-BERT$_{base}$ achieved a score of 74.00, while the original paper reported 76.25. Indeed, this difference comes from the different hyperparameters we adopt.

Different baselines adopt various configurations such as batch size, training precision (fp16 or fp32), and other factors. Recognizing that these factors significantly influence the final results, our aim is to assess different baselines under consistent conditions. To clarify, it would be misleading to evaluate, say, SimCSE-BERT$_{base}$ with a batch size of 64 while assessing SNCSE-BERT$_{base}$ with a batch size of 256. Such discrepancies could obscure the true reasons behind performance gaps. Therefore, we use a unified batch size of 256 for base models and 128 for large models.

To eliminate concerns about whether the proposed method can still work at baseline's optimal hyperparameters, we reassess the SimCSE-BERT$_{base}$ model in Table 10. Regardless of whether we use SimCSE's optimal settings or our uniform configuration, our method consistently outperforms the baseline.

Lastly, we want to mention that some baselines actually benefit from our standardized setup. For example, our reproduction of SimCSE-RoBERTa$_{base}$ saw an increase, going from the originally reported 76.57 to 77.09.

Table 10: A comparison between original SimCSE and reproduced SimCSE on seven semantic textual similarity (STS) datasets. Each method is evaluated on full test sets by Spearman's correlation, "all" setting.

| Method | batch | STS12 | STS13 | STS14 | STS15 | STS16 | STS-B | SICK-R | Avg. |
|---|---|---|---|---|---|---|---|---|---|
| Original SimCSE-BERT$_{base}$ | 64 | 68.40 | 82.41 | 74.38 | 80.91 | 78.56 | 76.85 | 72.23 | 76.25 |
| Reproduced SimCSE-BERT$_{base}$ | 256 | 66.92 | 78.44 | 71.15 | 79.50 | 77.71 | 75.50 | 68.76 | 74.00 |
| + HiCL | 256 | 69.04 | 80.68 | 72.71 | 80.39 | 78.68 | 76.96 | 70.35 | 75.54↑ |
| Reproduced SimCSE-BERT$_{base}$ | 64 | 69.17 | 82.02 | 73.41 | 80.54 | 78.36 | 76.60 | 71.96 | 76.01 |
| + HiCL | 64 | 69.44 | 82.10 | 74.48 | 81.62 | 78.77 | 77.75 | 70.26 | 76.35↑ |

## 6    Related Work

**Contrastive learning**   The recent ideas on contrastive learning originate from computer vision, where data augmentation techniques such as AUG-MIX (Hendrycks et al., 2020) and mechanisms like end-to-end (Chen et al., 2020a), memory bank (Wu et al., 2018), and momentum (He et al., 2020) have been tested for computing and memory efficiency. In NLP, since the success of SimCSE (Gao et al., 2021b), considerable progress has been made towards unsupervised sentence representation. This includes exploring text data augmentation techniques such as word repetition in ESimCSE (Wu et al., 2022b) for positive pair generation, randomly generated Gaussian noises in DCLR (Zhou et al., 2022), and negation of input in SNCSE (Wang et al., 2022) for generating negatives. Other approaches design auxiliary networks to assist contrastive objective (Chuang et al., 2022; Wu et al., 2022a). Recently, a combination of prompt tuning with contrastive learning has been developed in PromptBERT (Jiang et al., 2022) and SNCSE (Wang et al., 2022). With successful design of negative sample generation and utilization of prompt, SNCSE resulted in the state-of-the-art performance on the broadly-evaluated seven STS tasks. Our novel training paradigm, HiCL, is compatible with previous works and can be easily integrated with them. In our experiments, we re-train SimCSE, ESimCSE, and SNCSE with HiCL, showing a consistent improvement in all models.

**Hierarchical training**   The concept of hierarchical training has been proposed for long-text processing.   Self-attention models like Transformer (Vaswani et al., 2017) are limited by their input length, and truncation is their default method of processing text longer than the limit, which leads to information loss. To combat this issue, researchers have either designed hierarchical Transformers (Liu and Lapata, 2019; Nawrot et al., 2022) or adapted long inputs to fit existing Transform-ers (Zhang et al., 2019; Yang et al., 2020). Both solutions divide the long sequence into smaller parts, making full use of the whole input for increased robustness compared to those using partial information. Additionally, hierarchical training is usually more time efficient. SBERT (Reimers and Gurevych, 2019) employs a similar idea of hierarchical training. Instead of following traditional fine-tuning methods that concatenate two sentences into one for encoding in sentence-pair downstream tasks, SBERT found that separating the sentences and encoding them independently can drastically improve sentence embeddings. To our knowledge, we are the first to apply this hierarchical training technique to textual contrastive learning.

## 7    Conclusion

We introduce HiCL, the first hierarchical contrastive learning framework, highlighting the importance of incorporating local contrastive loss into the prior training paradigm. We delve into the optimal methodology for navigating the relationship between segments from same sequence in the computation of local contrastive loss. Despite the extra time required for slicing sequences into segments, HiCL significantly accelerates the encoding time of traditional contrastive learning models, especially for long input sequences. Moreover, HiCL excels in seamlessly integrating with various existing contrastive learning frameworks, enhancing their performance irrespective of their distinctive data augmentation techniques or foundational architectures. We employ SimCSE, ESimCSE, and SNCSE as case studies across seven STS tasks to demonstrate its scalability. Notably, our implementation with the SNCSE backbone model achieves the new state-of-the-art performance. This makes our hierarchical contrastive learning method a promising approach for further research in this area.

## Acknowledgment

The work is partly supported by the Endowment of Basic Sciences at the University of Michigan Medical School and by Dissertation Fellowships from the School of Information. Chaowei Xiao is supported by the U.S. Department of Homeland Security under Grant Award Number, 17STQAC00001-06-00.

We thank Fei Sun, Tianyu Gao, Simone Conia, Pratyush Maini, Amrith Setlur, Zhenhao Zhang, Jiazhao Li, members of the University of Michigan's NLP4Health research group, and all anonymous reviewers for helpful discussion and valuable feedback.

## Limitations

We acknowledge following limitations of our work:
1. Despite demonstrating the suitability of HiCL for pre-training scenarios in Section 5.5, we were not able to fully pre-train our model from scratch. Further research is needed to verify the effectiveness of HiCL in this context.
2. Some baselines (e.g., SimCSE) have shown that training a contrastive learning objective on human-labeled NLI datasets can lead to improved performance. We did not investigate this supervised setting.

## Ethics Statement

**Data**  All of the training data and evaluation benchmarks used in this paper are publicly available and do not pose any privacy concerns.

**AI Writing Assistance**  We have utilized the ChatGPT to polish our original content, rather than generate new ideas or suggestions. Despite the fact that EMNLP 2023 has not clearly stated whether the use of generative language models needs to be disclosed, we believe it is best to be transparent and explicitly disclose this.

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

## A Appendix

### A.1 Training Time

The practical training time can be complex, as the actual encoding time does not strictly follow a quadratic rule. However, our method demonstrates advantages in terms of efficiency when maintaining the same sequence truncation size. For example, while SimCSE-RoBERTa-large takes approximately 354.5 minutes for training, our method achieves the same task in just 152 minutes. The acceleration in time stems from two primary factors: 1) Savings in encoding time, as discussed in Section 3.3, and 2) The capability of HiCL to handle significantly larger batch sizes, enabling parallel processing for further acceleration. Please note that these advantages in efficiency are discussed specifically in the context of training with identical information, where the same truncation size is maintained. The original SimCSE, which adopts a truncation size of 32, is indeed faster than our approach as it does not need process truncated information.

| Backbone | $BERT_b$ | $BERT_l$ | $Roberta_b$ | $Roberta_l$ |
|---|---|---|---|---|
| SimCSE | 87.5 | 344.5 | 92.4 | 354.5 |
| + HiCL | 69.4 | 148.4 | 71.8 | 152.0 |

Table 12: Training minutes for models with 512 sequence truncation size. $_b$ for base models, $_l$ for large models.

### A.2 Transfer Learning

We also evaluate competing methods on several transfer tasks: MR (Pang and Lee, 2005), CR (Hu and Liu, 2004), SUBJ (Pang and Lee, 2004), MPQA (Wiebe et al., 2005), SST-2 (Socher et al., 2013), TREC (Voorhees and Tice, 2000), MRPC (Dolan and Brockett, 2005).

A comparison between SimCSE-BERT$_{base}$ (using its reported optimized hyperparameters and without adding MLM loss) and our model can be found in Table 11. While our approach may not demonstrate enhancements across all tasks, it does register improvements in 5 out of the 7 tasks, with each exceeding a 0.4% increase. The aggregate improvement stands at 0.39%.

### A.3 Experimental setup

Different baselines utilize varying training setups, including batch size, training precision (fp16 or fp32), and other factors. In order to maximally ensure a fair comparison, we unify the training setup across all competing methods and strictly follow each baseline's hyperparameter tuning process to re-tune the optimal hyperparameters accordingly.

Specifically, we employ a batch size of 256 for base models and 128 for large models. All base models are trained using full precision (fp32), while large models are trained using half precision (fp16) to mitigate potential memory issues with certain models. It is worth noting that disparities in performance between our re-run models and the original baselines may arise due to our intentional parameter adjustments to facilitate a direct comparison with other baselines.

Following the procedures used by SimCSE and SNCSE, we evaluate the model every 125 training steps on the development set of STS-B and select the best checkpoint for the final evaluation on the test sets.

### A.4 Hyperparameter tuning of HiCL

We first tune the weight of local contrastive $\alpha \in \{0.01, 0.05, 0.15\}$ along with learning rate 5e-6 for large models and 1e-5 for base models.[3] After fixing an optimized $\alpha$, we proceed to tune the learning rate using the suggested values for each model, including 5e-6 as an additional option. Specifically, we tune the learning rate to be one of the following values: {5e-6, 1e-5, 3e-5, 5e-5} for SimCSE models, {5e-6, 1e-5, 3e-5} for ESimCSE models, and {5e-6, 1e-5} for SNCSE models. The optmized hyperparameters of HiCL are listed in Table 13.

| Method | $\alpha$ | lr |
|---|---|---|
| SimCSE-BERT$_{base}$ | 0.15 | 5e-5 |
| SimCSE-BERT$_{large}$ | 0.15 | 3e-5 |
| SimCSE-RoBERTa$_{base}$ | 0.05 | 1e-5 |
| SimCSE-RoBERTa$_{large}$ | 0.05 | 5e-6 |
| ESimCSE-BERT$_{base}$ | 0.01 | 3e-5 |
| ESimCSE-BERT$_{large}$ | 0.01 | 5e-6 |
| ESimCSE-RoBERTa$_{base}$ | 0.01 | 1e-5 |
| ESimCSE-RoBERTa$_{large}$ | 0.01 | 5e-6 |
| SNCSE-BERT$_{base}$ | 0.01 | 1e-5 |
| SNCSE-BERT$_{large}$ | 0.01 | 1e-5 |
| SNCSE-RoBERTa$_{base}$ | 0.01 | 1e-5 |
| SNCSE-RoBERTa$_{large}$ | 0.15 | 5e-6 |

Table 13: Optimized hyperparameters of HiCL. lr: learning rate.

---

[3]We find that the results on the SICK-R development set were more consistent with the results on the seven STS test sets. Once an optimized checkpoint was identified through STS dev set, we fine-tuned the hyperparameters using the SICK-R development set.

Table 11: Transfer task results of sentence embedding performance (evaluted as accuracy). **Bold** marks the best result among competing methods under the same backbone model.

| Method | MR | CR | SUBJ | MPQA | SST2 | TREC | MRPC | Avg |
|---|---|---|---|---|---|---|---|---|
| SimCSE-BERT$_{base}$ | 80.34 | 85.35 | **94.65** | **89.21** | 84.68 | 88.00 | 73.10 | 85.05 |
| + HiCL | **80.76** | **86.39** | 94.61 | 89.02 | **85.17** | **88.60** | **73.68** | **85.46** |

For HiCLv2, we use the optimized values in HiCL and only tune for $\beta \in \{$1e-5, 3e-5, 1e-4, 3e-4$\}$. Optimized $\beta$ is shown in Table 14.

| Method | $\alpha$ | $\beta$ | lr |
|---|---|---|---|
| SimCSE-BERT$_{large}$ | 0.15 | 3e-4 | 5e-6 |
| SNCSE-BERT$_{large}$ | 0.01 | 3e-4 | 5e-6 |
| SimCSE-RoBERTa$_{large}$ | 0.05 | 3e-5 | 5e-6 |
| SNCSE-RoBERTa$_{large}$ | 0.15 | 1e-5 | 5e-6 |

Table 14: Optimized hyperparameters of HiCLv2 for four models in Section 5.7. lr: learning rate.

## A.5 Multiple Runs

In addition to the conventional practice of comparing models under the same random seed, we test the generalizability of our proposed method by using different random seeds, as presented in Table 15. An inherent challenge is the decision whether to retune the hyperparameters, considering the optimized ones under one initialization may vary under a different one. With the aim of investigating whether the optimized hyperparameters can be effectively transferred across various random seeds, we have opted not to retune the hyperparameters. [4]

HiCL improves performance across the identical nine backbone models as shown in Table 1, thereby demonstrating its robust generalization capabilities. The average scores in the multi-run setting are uniformly lower than those in the one-run setting, possibly due to the lack of sufficient hyperparameter tuning. This lack of tuning may also result in a reduced performance gap between HiCL and the baseline models.

## A.6 A Variant of HiCL

**Sequence-segment entailment.** The sequence-segment entailment objective is designed as an alternative way to model the relationship between

segments from the same sequence. Intuitively, segments from the same sequence are likely to be more similar to each other than to a random segment. However, this is not always the case, as segments from the same sequence can have contrary meaning. Modeling this relationship is difficult because it has both a high degree of correlation, yet no clear relationship between segments. To tackle this problem, we instead focus on modeling the relationship between a sequence and its segments. This is more straightforward, as we know that $seg_{i,j}$ comes from $seq_i$, and therefore they naturally form an entailment relationship. By doing this, we also retain the information about whether two segments come from the same sequence. Figure 3 provides an overview of this variant framework.

We employ a third contrastive objective to model the entailment relationship. Specifically, a segment $seg_{i,j}$ is entailed by sequence $seq_i$ but should not be entailed by $seq_k, \forall k \neq i$. Therefore, we treat $seg_{i,j}$ and $seq_i$ as a positive pair, and all other sequences in the batch as negative pairs with $seg_{i,j}$. We optimize the following InfoNCE loss function:

$$\mathcal{L}_e = -\log \frac{e^{sim(h_{i,j}, h_i)/\tau}}{e^{sim(h_{i,j}, h_i)/\tau} + \sum_{k \neq i} e^{sim(h_{i,j}, h_k)/\tau}}$$

(3)

**Overall objective of HiCLv2** The overall objective of HiCLv2 is a combination of local contrastive, global contrastive, and entailment loss, given by $\mathcal{L} = \alpha \mathcal{L}_l + \beta \mathcal{L}_e + (1 - \alpha - \beta)\mathcal{L}_g$, where $\alpha$ and $\beta$ are the weights.

## A.7 Computing Infrastructure

All models were trained on a single 48GB memory NVIDIA A40 GPU for one epoch, using the same initialization, i.e., the same random seed as used by all baselines, for one run. The server has the following configuration: Intel(R) Xeon(R) Gold 6226R CPU @ 2.90GHz x86-64 with CentOS 7 Linux operating system. PyTorch 1.7.1 is used as the programming framework.

---

[4]We note that both SimCSE-BERT-based models exhibit a substantial standard deviation, complicating the assessment of the models' performance. As a consequence, we only make minor adjustments to the learning rate for these two models for both baseline and HiCL method.

Table 15: Results of various contrastive learning methods on seven semantic textual similarity (STS) datasets. We report average results across 3 runs with different initialization. Each method is evaluated on full test sets by Spearman's correlation, "all" setting. **Bold** marks the best result among all competing methods under the same backbone model.

| Method | STS12 | STS13 | STS14 | STS15 | STS16 | STS-B | SICK-R | Avg. |
|---|---|---|---|---|---|---|---|---|
| SimCSE-BERT$_{base}$ | $67.41_{\pm1.2}$ | $79.63_{\pm2.4}$ | $72.06_{\pm2.7}$ | $79.71_{\pm2.0}$ | $77.77_{\pm1.0}$ | $75.44_{\pm2.2}$ | $69.69_{\pm1.2}$ | $74.53_{\pm1.8}$ |
| + HiCL | $67.04_{\pm1.7}$ | $80.94_{\pm0.3}$ | $71.93_{\pm0.7}$ | $80.07_{\pm0.5}$ | $78.16_{\pm0.5}$ | $75.55_{\pm1.2}$ | $69.57_{\pm1.1}$ | $74.75_{\pm0.7}\uparrow$ |
| ESimCSE-BERT$_{base}$ | $70.07_{\pm1.9}$ | $81.95_{\pm1.8}$ | $75.35_{\pm1.9}$ | $82.85_{\pm0.6}$ | $79.44_{\pm0.8}$ | $79.66_{\pm0.7}$ | $70.93_{\pm0.3}$ | $77.18_{\pm1.1}$ |
| + HiCL | $70.24_{\pm0.3}$ | $82.34_{\pm0.8}$ | $76.05_{\pm0.6}$ | $83.03_{\pm0.2}$ | $79.73_{\pm0.5}$ | $79.47_{\pm0.6}$ | $71.40_{\pm1.1}$ | $77.46_{\pm0.4}\uparrow$ |
| SNCSE-BERT$_{base}$ | $70.53_{\pm0.4}$ | $84.37_{\pm0.1}$ | $\mathbf{76.95}_{\pm0.1}$ | $\mathbf{83.59}_{\pm0.4}$ | $\mathbf{80.28}_{\pm0.1}$ | $\mathbf{81.28}_{\pm0.2}$ | $\mathbf{74.99}_{\pm0.1}$ | $\mathbf{78.86}_{\pm0.1}$ |
| + HiCL | $\mathbf{70.87}_{\pm0.3}$ | $\mathbf{84.48}_{\pm0.2}$ | $76.84_{\pm0.2}$ | $83.21_{\pm0.2}$ | $79.61_{\pm0.3}$ | $80.87_{\pm0.3}$ | $74.85_{\pm0.2}$ | $78.67_{\pm0.1}$ |
| SimCSE-BERT$_{large}$ | $70.03_{\pm0.9}$ | $82.55_{\pm1.2}$ | $74.34_{\pm1.6}$ | $82.96_{\pm0.9}$ | $77.86_{\pm0.9}$ | $79.09_{\pm1.2}$ | $70.44_{\pm4.4}$ | $76.75_{\pm0.9}$ |
| + HiCL | $68.38_{\pm2.4}$ | $83.03_{\pm2.2}$ | $74.04_{\pm1.4}$ | $83.10_{\pm0.9}$ | $78.22_{\pm1.0}$ | $77.89_{\pm2.0}$ | $73.99_{\pm0.3}$ | $76.95_{\pm1.4}\uparrow$ |
| ESimCSE-BERT$_{large}$ | $72.25_{\pm0.9}$ | $84.77_{\pm0.8}$ | $76.69_{\pm0.8}$ | $84.38_{\pm0.6}$ | $79.40_{\pm0.6}$ | $80.36_{\pm0.7}$ | $74.37_{\pm0.7}$ | $78.89_{\pm0.7}$ |
| + HiCL | $\mathbf{73.24}_{\pm0.8}$ | $84.93_{\pm0.5}$ | $77.24_{\pm0.1}$ | $84.73_{\pm0.1}$ | $80.08_{\pm0.3}$ | $80.79_{\pm0.2}$ | $74.23_{\pm0.3}$ | $79.32_{\pm0.1}\uparrow$ |
| SNCSE-BERT$_{large}$ | $71.76_{\pm0.5}$ | $86.59_{\pm0.3}$ | $\mathbf{78.60}_{\pm0.3}$ | $\mathbf{85.61}_{\pm0.3}$ | $80.46_{\pm0.2}$ | $82.07_{\pm0.3}$ | $75.23_{\pm0.8}$ | $80.05_{\pm0.3}$ |
| + HiCL | $72.34_{\pm1.0}$ | $\mathbf{86.61}_{\pm0.3}$ | $78.52_{\pm0.1}$ | $85.43_{\pm0.2}$ | $\mathbf{80.47}_{\pm0.4}$ | $\mathbf{82.14}_{\pm0.3}$ | $\mathbf{75.43}_{\pm0.5}$ | $\mathbf{80.13}_{\pm0.3}\uparrow$ |
| SimCSE-RoBERTa$_{base}$ | $69.65_{\pm0.3}$ | $81.86_{\pm0.4}$ | $74.19_{\pm0.7}$ | $82.46_{\pm0.7}$ | $81.40_{\pm0.3}$ | $81.17_{\pm0.4}$ | $68.88_{\pm0.7}$ | $77.08_{\pm0.5}$ |
| + HiCL | $69.02_{\pm0.6}$ | $81.94_{\pm0.4}$ | $74.05_{\pm0.5}$ | $82.66_{\pm0.6}$ | $\mathbf{81.62}_{\pm0.7}$ | $81.20_{\pm0.6}$ | $68.87_{\pm0.7}$ | $77.05_{\pm0.5}$ |
| ESimCSE-RoBERTa$_{base}$ | $69.49_{\pm0.5}$ | $81.42_{\pm0.7}$ | $73.76_{\pm0.4}$ | $81.88_{\pm0.6}$ | $80.98_{\pm0.4}$ | $80.62_{\pm0.3}$ | $68.74_{\pm0.6}$ | $76.70_{\pm0.4}$ |
| + HiCL | $69.73_{\pm0.2}$ | $81.48_{\pm0.2}$ | $74.35_{\pm0.3}$ | $82.61_{\pm0.3}$ | $81.23_{\pm0.4}$ | $80.67_{\pm0.2}$ | $68.69_{\pm0.6}$ | $76.97_{\pm0.2}\uparrow$ |
| SNCSE-RoBERTa$_{base}$ | $\mathbf{70.47}_{\pm0.2}$ | $83.27_{\pm0.4}$ | $\mathbf{76.15}_{\pm0.4}$ | $\mathbf{84.13}_{\pm0.1}$ | $81.48_{\pm0.7}$ | $\mathbf{81.83}_{\pm0.5}$ | $\mathbf{73.18}_{\pm0.9}$ | $\mathbf{78.65}_{\pm0.3}$ |
| + HiCL | $70.16_{\pm0.9}$ | $\mathbf{83.41}_{\pm0.3}$ | $75.73_{\pm0.3}$ | $83.90_{\pm0.2}$ | $81.32_{\pm0.5}$ | $81.36_{\pm0.6}$ | $72.40_{\pm1.0}$ | $78.33_{\pm0.5}$ |
| SimCSE-RoBERTa$_{large}$ | $71.13_{\pm0.8}$ | $83.33_{\pm0.5}$ | $75.69_{\pm0.3}$ | $84.04_{\pm0.5}$ | $80.74_{\pm0.5}$ | $81.97_{\pm0.4}$ | $71.13_{\pm0.3}$ | $78.29_{\pm0.2}$ |
| + HiCL | $71.48_{\pm1.0}$ | $83.84_{\pm0.6}$ | $75.90_{\pm0.4}$ | $84.86_{\pm0.4}$ | $81.91_{\pm0.5}$ | $82.33_{\pm0.6}$ | $71.62_{\pm0.2}$ | $78.85_{\pm0.4}\uparrow$ |
| ESimCSE-RoBERTa$_{large}$ | $72.70_{\pm1.5}$ | $83.49_{\pm0.2}$ | $76.39_{\pm0.7}$ | $83.07_{\pm1.1}$ | $80.53_{\pm0.3}$ | $82.27_{\pm0.5}$ | $72.66_{\pm0.6}$ | $78.73_{\pm0.7}$ |
| + HiCL | $72.46_{\pm0.7}$ | $83.74_{\pm0.6}$ | $76.29_{\pm0.4}$ | $84.56_{\pm0.0}$ | $81.20_{\pm0.2}$ | $82.89_{\pm0.6}$ | $71.58_{\pm0.7}$ | $78.98_{\pm0.4}\uparrow$ |
| SNCSE-RoBERTa$_{large}$ | $\mathbf{73.18}_{\pm0.5}$ | $\mathbf{86.15}_{\pm0.5}$ | $\mathbf{79.08}_{\pm0.3}$ | $\mathbf{86.81}_{\pm0.3}$ | $82.40_{\pm0.1}$ | $83.48_{\pm0.4}$ | $77.15_{\pm0.4}$ | $81.18_{\pm0.2}$ |
| + HiCL | $72.99_{\pm0.7}$ | $85.74_{\pm0.6}$ | $79.07_{\pm0.7}$ | $86.45_{\pm0.2}$ | $\mathbf{82.77}_{\pm0.4}$ | $\mathbf{83.70}_{\pm0.7}$ | $\mathbf{77.84}_{\pm0.5}$ | $\mathbf{81.22}_{\pm0.5}\uparrow$ |

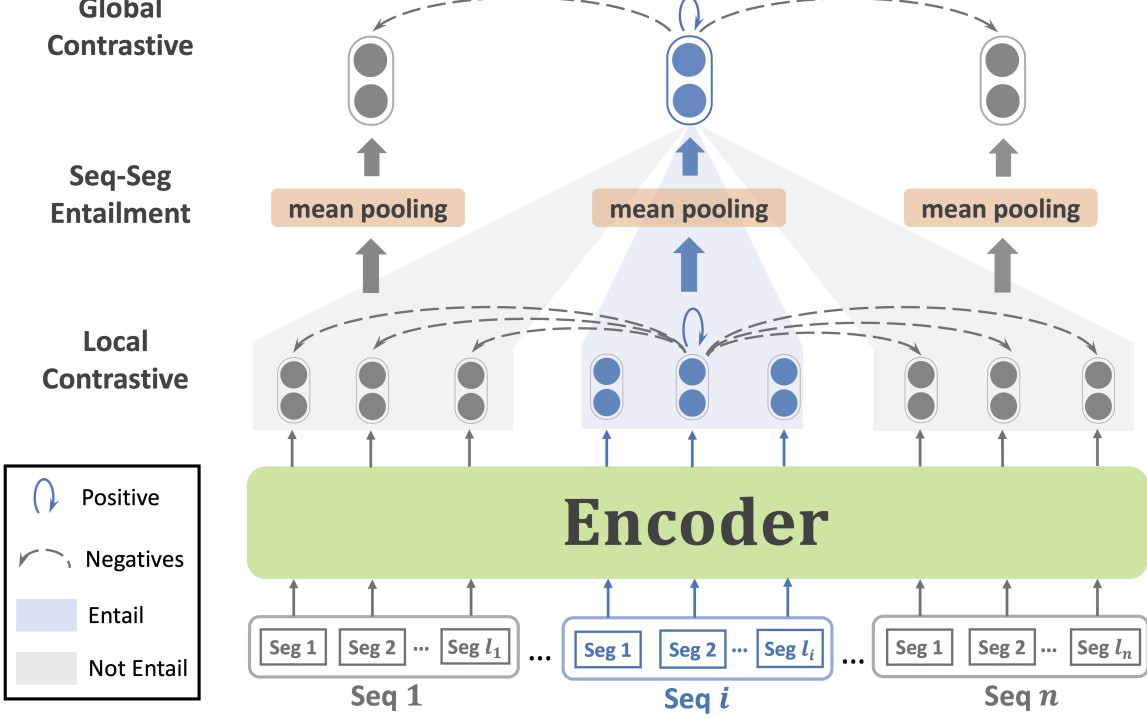

Figure 3: The overview of HiCLv2 framework with local contrastive, global contrastive, and sequence-segment entailment objective.

Table 16: Results of various contrastive learning methods on seven semantic textual similarity (STS) datasets. We report average results across 3 runs with different initialization. Each method is evaluated on full test sets by Spearman's correlation, "all" setting.

| Method | loss | STS12 | STS13 | STS14 | STS15 | STS16 | STS-B | SICK-R | Avg. |
|---|---|---|---|---|---|---|---|---|---|
| SimCSE+HiCL | summation | $71.48_{\pm1.0}$ | $83.84_{\pm0.6}$ | $75.90_{\pm0.4}$ | $84.86_{\pm0.4}$ | $81.91_{\pm0.5}$ | $82.33_{\pm0.6}$ | $71.62_{\pm0.2}$ | $78.85_{\pm0.4}\uparrow$ |
| SimCSE+HiCL | averaging | $71.35_{\pm0.5}$ | $83.86_{\pm0.2}$ | $75.71_{\pm0.1}$ | $84.73_{\pm0.3}$ | $82.00_{\pm0.3}$ | $82.10_{\pm0.1}$ | $71.11_{\pm0.2}$ | $78.70_{\pm0.1}$ |
| ESimCSE+HiCL | summation | $72.46_{\pm0.7}$ | $83.74_{\pm0.6}$ | $76.29_{\pm0.4}$ | $84.56_{\pm0.0}$ | $81.20_{\pm0.2}$ | $82.89_{\pm0.6}$ | $71.58_{\pm0.7}$ | $78.98_{\pm0.4}\uparrow$ |
| ESimCSE+HiCL | averaging | $71.81_{\pm0.3}$ | $83.53_{\pm0.3}$ | $75.77_{\pm0.3}$ | $84.63_{\pm0.2}$ | $81.60_{\pm0.1}$ | $82.45_{\pm0.1}$ | $71.32_{\pm0.3}$ | $78.73_{\pm0.1}$ |
| SNCSE+HiCL | summation | $72.99_{\pm0.7}$ | $85.74_{\pm0.6}$ | $79.07_{\pm0.7}$ | $86.45_{\pm0.2}$ | $82.77_{\pm0.4}$ | $83.70_{\pm0.7}$ | $77.84_{\pm0.5}$ | $81.22_{\pm0.5}\uparrow$ |
| SNCSE+HiCL | averaging | $72.79_{\pm0.9}$ | $86.05_{\pm0.2}$ | $79.12_{\pm0.3}$ | $86.47_{\pm0.4}$ | $83.03_{\pm0.2}$ | $83.53_{\pm0.3}$ | $77.10_{\pm0.9}$ | $81.16_{\pm0.3}$ |

## A.8 Summation or Averaging in Contrasting Negatives

In Eq. 2, we contrast a segment $h_{i,j}$ with segments from different sequences $h_{k,*}$ where $k \neq i$, which is more like picking the correct pair out of a very big batch. We investigate an alternative way which includes a weighted factor when computing similarities between $h_{i,j}$ and $h_{k,*}$. Specifically, $\sum_{k \neq i} w_k \times e^{sim(h_{i,j}, h_{k,*})/\tau}$ assumes contrasting over a fixed small batch instead of using an extended big batch, where $w_k = \frac{1}{v}$, $v$ is the number of segments that sequence k is divided into. In essence, this design would allocate varying weights to different segments in computation: segments derived from shorter sequences would receive a higher weight, while those from longer sequences would be assigned a lower weight. A critical query here is: should we differentiate the importance of segments based on their originating sequence length? To answer this question, we compare this new design (named *averaging*) with our previous version in Eq. 2 (named *summation*).

We benchmark the *averaging* loss against *summation* using three different models: SimCSE, ESimCSE, and SNCSE, all on the RoBERTa$_{large}$ backbone. To control for initialization effects, we've provided results from three-run evaluations on seven STS datasets in Table 16. It is clear that the *summation* loss consistently outperforms the *averaging* loss, which proves that varying the importance of segments based on source length may not yield beneficial outcomes.