# OpenReview forum: "HiCL: Hierarchical Contrastive Learning of Unsupervised Sentence Embeddings"
_EMNLP/2023/Conference — EMNLP 2023 Findings_

### Official Review · Reviewer_pM2M · 2023-08-03

**Typos Grammar Style And Presentation Improvements:** N/A
**Soundness:** 2

**Excitement:**

2: Mediocre: This paper makes marginal contributions (vs non-contemporaneous work), so I would rather not see it in the conference.

**Justification For Ethical Concerns:**

Justification for Ethical Concerns


**Missing References:**

N/A

**Paper Topic And Main Contributions:**

The objective of the study is to design a hierarchical contrastive learning framework by considering the local segment-level and global sequence-level relationships. Given a sentence, the proposed model called HiCL first divides it into smaller segments and encodes each segment to calculate local segment representation. Then, it gets the global sequence representation by aggregating the local segment representations. In addition, HiCL can reduce the training time by first encoding short segments and then aggregating them to obtain the sequence representation. For the contrastive learning, they use the dropout of SimCSE to get a positive pair and use the negated sentences and prompt tuning of SNCSE as negative examples. The experimental results show that HiCL outperforms existing contrastive learning methods such as SimCSE, ESimCSE, and SNCSE.

**Questions For The Authors:**

- In p.228, the authors mentioned that most input instances can be divided into three or fewer segments. This means that the sequence length is usually not long. If most of the inputs are long sequences, it makes sense to divide the sequence to obtain a segment-level representation, but in other cases, it is difficult to expect a significant effect of using segment-level representation.
- This paper is assuming that the length of segments in a sequence is the same. However, the reviewer thinks that each segment length should be variable according to its locality. For example, given a sentence like “I like a banana but hate an apple”, if we assume that the number of segments is 3 as the segment length, the segments are <I like a>, <banana but hate>, and <an apple>. In this case, each segment does not contain its meaning well. In this example, is it correct that the segments are <I like a banana> and <but hate an apple>?
- In Equation 2, is the term of the denominator about negative segments (i.e., sigma_{k<>i} sigma_{v} e^{sim(h_{i,j}, h_{k,v})/tau}) correct? I think that averaging e^{sim(h_{i,j}, h_{k,v})/tau} is better than the summation.
- To see the robustness of the proposed model, it is recommended to apply it to other natural language processing tasks as well as the semantic textual similarity task.

**Reasons To Accept:**

- This paper propose a new hierarchical contrastive learning framework that considers local segment-level relationships, unlike existing methods, and experimented it with seven semantic textual similarity data sets, showing better performance than existing methods such as SimCSE, ESimCSE, and SNCSE.
- This paper presents that the encoding process of the proposed method can reduce training time rather than prior contrastive learning models. Compared to the existing methods, the experimental results show that the encoding time of the proposed method is reduced, and the time complexity is theoretically analyzed.

**Reasons To Reject:**

- Although the basic idea of this paper can be understood, the paper needs to be partially updated for readability. It is not clear how the encoder in Figure 1 is added on BERT and RoBERTa. Is it correct to use the proposed hierarchical contrastive leaning as a loss function for BERT or RoBERTa model? What does the arrows in Figure 1 mean? It is ambiguous whether it simply means a positive and negative relationship, or whether it affects finding negative samples.
- This reviewer is not sure what the research goal is in the paper. Is the final goal to do sentence embedding or to solve a specific natural language processing problem such as semantic textual similarity through the proposed hierarchical contrastive learning?
- Excepting considering the segment-level relationship, it is not sure what else the technical contribution is in the proposed method.

**Reproducibility:**

3: Could reproduce the results with some difficulty. The settings of parameters are underspecified or subjectively determined; the training/evaluation data are not widely available.

**Reviewer Confidence:**

3: Pretty sure, but there's a chance I missed something. Although I have a good feel for this area in general, I did not carefully check the paper's details, e.g., the math, experimental design, or novelty.

---

> ### Author Rebuttal · Authors · 2023-08-29
>
> We thank the reviewer for their insightful feedback. We address the comments below.
>
> > *1. Unclear figure*
>
> Thanks for pointing out that our current figure makes some confusion to your interpretation. Here we make some clarifications:
>
> 1. "It is not clear how the encoder in Figure 1 is added on BERT and RoBERTa"
>
> * The encoder itself is a BERT or RoBERTa.
>
> 2. "Is it correct to use the proposed hierarchical contrastive leaning as a loss function for BERT or RoBERTa model?"
>
> * Yes, it is. The BERT and RoBERTa only serve as the basic backbone model to extract text features initially. SimCSE[1] also uses BERT and RoBERTa as the backbone, and computes contrastive loss based on the extracted sentence features.
>
> 3. "What does the arrows in Figure 1 mean?"
>
> * In Figure 1, the green dashed arrows indicate that the destination vector serves as a negative instance in relation to the origin vector. Conversely, the blue arrows depict two passes of the same text with varying dropout, forming a positive pair. This is consistent with SimCSE's definition.
>
> We wish to emphasize that our visual depiction of the training process is inspired by well-established conventions. A comparable illustration can be seen in Figure 1 of the SimCSE paper. However, we will refine the figure for clarity in our final version.
>
> > *2. Research goal*
>
> In essence, we introduce a new contrastive learning paradigm that enhances existing CL frameworks, boosting both encoding efficiency and effectiveness to learn a good representation. That is to say. We are not aiming to design for the specific task such as semantic textual similarity but with the goal to learn a good representation.
> However, while the majority of preceding studies [1,2,3,4,5] focus their evaluation of CL objectives solely on the STS benchmark, SimCSE[1] emphasizes that for effective performance on other transfer NLP tasks, the CL objective must be complemented with additional losses. Given our mission to enhance the CL training paradigm, we've chosen to present our results on the STS benchmark, staying consistent with the prevalent norms in the CL community.
> If you believe that prevailing conventions within this community have led to misunderstandings, we are open to refining our assertions in the revised version.
>
> > *3. Other technique contribution than segment contrastive learning*
>
> Segment-level contrastive learning is our center technique contribution. We made a lot of effort to explore what’s the most proper way to make it work, including but not limited to:
> 1. studying how to aggregate segment representation to get sequence representation;
> 2. figuring how to define the relationship between segments from the same sequence, shall we treat them as positive pairs or negative pairs or neither;
> 3. finding optimal truncation length;
> 4. studying the proper weight between two losses
> 5. exploring a variant of HiCL that adds entailment judgment between segments and sequences.
>
> We believe all these efforts together make our technique contribution strong, as also recognized by other reviewers:
> 1. Reviewer etFB: "The paper presents a **pioneering** approach"
> 2. Reviewer 69hi: "First to propose a hierarchical contrastive learning framework (which is a **bold claim** and **surprising** if true)"
>
> *Q1. Meaning of local contrastive when input is short*
>
> We agree with you that our method becomes particularly compelling when handling a majority of long inputs. As mentioned in Section 5.4, HiCL shows more improvements on a long training corpus (WikiText-103 with 3617 tokens per article).
> Prior studies [1,2,3,4,5] uniformly trained on a short-input dataset, thus, to ensure a fair comparison, we opted to utilize the same data for our primary evaluation. Importantly, the impressive performance on this short dataset happens to demonstrate the superiority of our method, given its adaptability even with fewer segments.
>
> *Q2. Issue with fixed truncation*
>
> This is a good point, we actually have also observed the issue with fixed truncation size in Section 5.4 through the experiment that trains a solo local contrastive model.
> Better segmentation could lead to a better performance, but will also come with other issues such as low encoding efficiency as it slices into more segments compared to fixed truncation strategy.
> Further, traditional methods also introduce noise into training. Take your example “I like a banana but hate an apple” with truncation size 3 for illustration, traditional CL only contrasts using “I like a”, which can also mislead the model.
> Overall, even with our current fixed truncation strategy, we have already attained commendable results and plan to leave the exploration of dynamic truncation as our future work.
>
> *Q3. Suggestion on revised loss*
>
> Thanks for your suggestion. Our current version contrasts a segment ($h_{i,j}$) with every other segment from different sequences ($h_{k, v}$ where $k!= i$), which is more like picking the correct pair out of a big batch.
> We interpret you were suggesting to add a weighted average factor in computing $\sum_{k!=i}(w_k \times \sum_{v}e^{sim(h_{i,j}, h_{k,v})/\tau}$), where the weight $w_k$ can be defined as the inverse of #segments that sequence k is divided into, i.e., $w_k = \frac{1}{\sum_{v} 1}$. In essence, this would allocate varying weights to different segments in computation: segments derived from shorter sequences would receive a higher weight, while those from longer sequences would be assigned a lower weight.
>
> A critical query here is: **why should we differentiate the importance of segments based on their originating sequence length?** Our approach treats each segment as an independent contrastive objective, and we don't think that varying their importance based on source length would be beneficial.
>
> *Q4. Recommendation on more evaluation tasks*
>
> Our primary objective is to refine existing CL techniques, focusing on both encoding efficiency and performance. To ensure fairness and comparability, we've adhered to evaluation standards set by prior research. Esteemed publications like [1,2,3,4], along with the recent miCSE [5] from ACL '23, consistently employ the benchmark we've chosen.
> Thanks for your recommendation, we are committed to broadening our evaluation scope in the upcoming revision, incorporating transfer tasks: MR, CR, SUBJ, MPQA, SST-2, TREC, and MRPC.
>
> -----
> [1] Gao, Tianyu, Xingcheng Yao, and Danqi Chen. "Simcse: Simple contrastive learning of sentence embeddings." in EMNLP 2021
>
> [2] Wu, Xing, Chaochen Gao, Liangjun Zang, Jizhong Han, Zhongyuan Wang, and Songlin Hu. "Esimcse: Enhanced sample building method for contrastive learning of unsupervised sentence embedding." in COLING 2022
>
> [3] Wang, Hao, and Yong Dou. "SNCSE: contrastive learning for unsupervised sentence embedding with soft negative samples." In International Conference on Intelligent Computing, pp. 419-431. Singapore: Springer Nature Singapore, 2023.
>
> [4] Zhou, Kun, Beichen Zhang, Wayne Xin Zhao, and Ji-Rong Wen. "Debiased contrastive learning of unsupervised sentence representations." in ACL 2022
>
> [5] Klein, Tassilo, and Moin Nabi. "miCSE: Mutual Information Contrastive Learning for Low-shot Sentence Embeddings." in ACL 2023

---

### Official Review · Reviewer_etFb · 2023-08-04

**Soundness:** 4

**Excitement:**

3: Ambivalent: It has merits (e.g., it reports state-of-the-art results, the idea is nice), but there are key weaknesses (e.g., it describes incremental work), and it can significantly benefit from another round of revision. However, I won't object to accepting it if my co-reviewers champion it.

**Paper Topic And Main Contributions:**

The paper introduces a novel hierarchical contrastive learning framework (referred to as HiCL) that incorporates both local (segment-level) and global (sentence-level) relationships for unsupervised sentence embedding. The experimental findings demonstrate the efficacy of the proposed framework, consistently outperforming baseline models in the majority of cases. Furthermore, the framework showcases notable efficiency by significantly reducing training time.

**Questions For The Authors:**

1. Does the author solely focus on linearly splitting sentences on average? The exploration of the hierarchical structure and semantics of the segments seems limited, which I believe is crucial for constructing positive and negative samples.
2. Segment-level encoding only takes into account tokens within each segment. Could this potentially eliminate long-context dependencies, such as when two semantically related words in the same sentence are divided into separate segments?
3. Does considering segments from two sentences solely as negatives introduce noise or even incorrect samples? If so, how can this be avoided? Providing examples in the paper would enhance understanding.

**Reasons To Accept:**

1. The paper presents a pioneering approach by incorporating hierarchical relationships into contrastive learning within the field of NLP, as explicitly mentioned. While the application of hierarchical training in transformer-like architectures is well-documented in the second part of the Related Work section, the introduction of local contrastive objectives is a novel contribution.
2. The paper has a comprehensive intrinsic analysis, and it is well-written and easy for readers to follow.

**Reasons To Reject:**

The motivation provided in the paper to apply contrastive learning on shorter segments may not be sufficiently strong. The efficiency aspect mentioned on line 54, which highlights that hierarchical training is more time-saving due to encoding shorter contexts, seems unrelated to contrastive learning itself. In other words, there exist other efficient methods using BERT that can also reduce computational costs.

In terms of effectiveness (lines 54-61), the paper argues against solely relying on using entire sentences for contrastive learning. However, it is questionable whether using shorter segments alone is adequate, as focusing solely on sentence length may overlook other important factors. For example, shorter segments might introduce more noise and make it easier for two segments (from different sentences) to appear identical, resulting in them being mistakenly labeled as negative samples.

This suggests that a more comprehensive consideration of various factors beyond just segment length is necessary when exploring the effectiveness of contrastive learning.

**Reproducibility:**

4: Could mostly reproduce the results, but there may be some variation because of sample variance or minor variations in their interpretation of the protocol or method.

**Reviewer Confidence:**

3: Pretty sure, but there's a chance I missed something. Although I have a good feel for this area in general, I did not carefully check the paper's details, e.g., the math, experimental design, or novelty.

**Typos Grammar Style And Presentation Improvements:**

L250 "also not" -> "not ... either"

L569 "same sequence" -> "the same sequence"

---

> ### Author Rebuttal · Authors · 2023-08-29
>
> We thank the reviewer for their constructive feedback. We will fix the typo pointed out. We address the questions in below:
>
> > *1. Motivation for local segments*
>
> In general, the local contrastive is proposed for:
>
> * Making the task of distinguishing positives and negatives more challenging (stated between Line 54-61) for learning better representation. More specifically, encoding as a whole makes contrasting task too easy to learn meaningful representations, given that longer sequences naturally convey more information. As highlighted in Lines 58-61, CLEAR[1] has shown that naive sequence-level augmentation can lead to the failure of CL training. Similarly, SNCSE[2]  concluded that introducing hard negatives increases the challenge of identifying positive pairs, ultimately leading to a more effective model.
> * Processing longer sequences (stated between Line 66-73) and avoiding information loss by not dropping the part exceeds length limitation.
> * Addressing the expensive cost for encoding the entire sequence directly (stated between Line 52-54).
>
> Therefore, to enhance encoding efficiency and intensify the challenges of the contrastive task, we opted for including local contrastive loss.
>
> > *2. Potential noise introduced by fixed slicing length*
>
> This is a good point, we actually have also observed the issue with fixed truncation size in Section 5.4 through the experiment that trains a solo local contrastive model.
> Better segmentation could lead to a better performance, but will also come with other issues such as low encoding efficiency as it slices into more segments compared to fixed truncation strategy.
> Further, traditional methods also introduce noise into training. For instance, “I like a banana but hate an apple” with truncation size 3 for illustration, traditional CL only contrasts using “I like a”, which can also mislead the model.
> Overall, even with our current fixed truncation strategy, we have already attained commendable results and plan to leave the exploration of dynamic truncation for future work.
>
> *Q1. Exploration on other hierarchical structure and semantics of the segments*
>
> We acknowledge that other slicing ways (such as truncation by punctuation) might bring more improvements. However, as we mentioned between Line 419-425, due to time constraints and consideration for encoding efficiency, we have opted for a fixed-length truncation approach and leave the investigation of the approaches you mentioned to future work.
>
> *Q2. Drawback of segment-level encoding*
>
> We agree that segment-level encoding could eliminate some long-text dependency. Yet, we compensate it somehow via aggregating all segment information to form the whole sequence-level representation.
> It's worth noting that this isn't a challenge exclusive to our method. Traditional CL also meets with this issue when inputs surpass its length constraint, resorting to a break of long-text dependency.
>
> *Q3. The relationship of two segments from different sequences*
>
> At present, we regard segments from disparate sequences as negative pairs. We recognize the potential pitfall of this approach, as it could give rise to false negatives—segments from distinct sequences could be semantic similar.
> Nonetheless, this challenge isn't unique to our method. Traditional CL also involves false negatives when designating all other instances in a batch as negatives.
>
> We noticed that one paper DCLR[3] partially addresses this issue by re-assigning the negative pairs that have a high semantic similarity (larger than a threshold). However, its computing of semantic similarity of each pair requires an external model, which makes it much more computationally expensive.
>
> -----
> [1] Wu, Zhuofeng, Sinong Wang, Jiatao Gu, Madian Khabsa, Fei Sun, and Hao Ma. "Clear: Contrastive learning for sentence representation." arXiv preprint arXiv:2012.15466 (2020).
>
> [2] Wang, Hao, and Yong Dou. "SNCSE: contrastive learning for unsupervised sentence embedding with soft negative samples." In International Conference on Intelligent Computing, pp. 419-431. Singapore: Springer Nature Singapore, 2023.
>
> [3] Zhou, Kun, Beichen Zhang, Wayne Xin Zhao, and Ji-Rong Wen. "Debiased contrastive learning of unsupervised sentence representations." in ACL 2022

---

### Official Review · Reviewer_2q7u · 2023-08-08

**Soundness:** 3

**Excitement:**

3: Ambivalent: It has merits (e.g., it reports state-of-the-art results, the idea is nice), but there are key weaknesses (e.g., it describes incremental work), and it can significantly benefit from another round of revision. However, I won't object to accepting it if my co-reviewers champion it.

**Paper Topic And Main Contributions:**

The paper proposes using hierarchical contrastive learning to improve unsupersvised sentence representation learning. The encoding efficiency of the method might be beneficial to the field.

**Questions For The Authors:**

see reason to reject (especially better motivation for the central method).

**Reasons To Accept:**

1. The "intrinsic study and discussion" section is thorough and sound.
2. The encoding time advantage might be useful when generalizing to datasets with longer documents.

**Reasons To Reject:**

1. The local contrastive loss is not well-motivated. It can be questioned that this is just beneficial/relavant to the optimization of the training. Empirically, this trick might be similar to baseline methods + a larger batch size.

2. The performance improvement is very small. Also, in the main table, the reproduction of SimCSE BERT base is over 2 absolute points lower than reported in the original paper, and the reproduction of BERT large also falls largely behind perception from people with experience in the field (when you check Table 12 when the authors did 3 runs, this seems to be relatively fixed, but making the performance improvement even smaller).

3. I will strongly suggest that the authors prove the effectiveness of this method on generalizing to longer documents, instead of competing on sts tasks. The authors first claim that baseline methods might fail to generalize to shorter sequence, but wikipedia-1M (train) and sts (test) are both short, making sts not a good suite to test out the superiority of the method (in terms of length generalization). In 3.2, the authors state that "By enabling the model to focus on short sentences, local contrastive learning allows the model to better match the sentence length distribution, as longer sentences are less common." If the authors understand this correctly, it'll be more sound to prove the effectiveness of the method on train-test length mismatch settings (train on long, test on short; train on short, test on long).

**Reproducibility:**

3: Could reproduce the results with some difficulty. The settings of parameters are underspecified or subjectively determined; the training/evaluation data are not widely available.

**Reviewer Confidence:**

4: Quite sure. I tried to check the important points carefully. It's unlikely, though conceivable, that I missed something that should affect my ratings.

---

> ### Author Rebuttal · Authors · 2023-08-29
>
> We thank the reviewer for their helpful feedback. We address the comments below.
>
> > *1. Not well-motivated for local contrastive loss* and *the benefits of our method*
>
> > *1.1 Not well-motivated for local contrastive loss*
>
> In general, the local contrastive is proposed for:
> * **Making the task of distinguishing positives and negatives more challenging** (stated between Line 54-61) for learning better representation. More specifically, encoding as a whole makes contrasting task too easy to learn meaningful representations, given that longer sequences naturally convey more information. As highlighted in Lines 58-61, CLEAR[1] has shown that naive sequence-level augmentation can lead to the failure of CL training. Similarly, SNCSE[2] concluded that introducing hard negatives increases the challenge of identifying positive pairs, ultimately leading to a more effective model.
> * **Processing longer sequences** (stated between Line 66-73) and **avoiding information loss** by not dropping the part exceeds length limitation.
> * **Addressing the expensive cost for encoding** the entire sequence directly (stated between Line 52-54).
>
> Therefore, to enhance encoding efficiency and intensify the challenges of the contrastive task, we opted for including local contrastive loss.
>
> > *1.2 The benefits of our method*
>
> To address your concern regarding whether our proposed method simply serves as an alternative to the baseline with a larger batch size, we have undertaken further experiments. Given that in 99.9% of cases the training data can be segmented into four or fewer segments, we conducted a comparison between the baseline (based on SimCSE-BERT_base) with a batch size 4 times larger than our method. The results are as follows:
>
> |        |  batch  |  STS12 |   STS13    |STS14| STS15  | STS16 | STS-B |  SICK-R |   Avg. |
> |:-----:|:--------:|:----------:|:------------:|:--------:|:-----:|:--------:|:----------:|:------------:|:--------:|
> | SimCSE| 64     | 69.17   | 82.02  | 73.41  | 80.54 |78.36 |76.60  | 71.96   |76.01|
> | SimCSE| 256   | 66.92  | 78.44  | 71.15  | 79.50 | 77.71 | 75.50 | 68.76  | 74.00 |
> |+HiCL     | 64     | 69.44 | 82.10    | 74.48 | 81.62 | 78.77 | 77.75   | 70.26   | 76.35↑|
>
> From above table:
> * **Clearly, SimCSE, when using a batch size of 256, lags behind our model that operates with a smaller batch size of 64 (74.00 vs. 76.35)**.
> * Increasing the batch size alone isn't a surefire way to boost performance, as corroborated by the SimCSE paper.
>
> Further, **scaling up baselines by simply ramping up the batch size introduces computational challenges**. For instance, attempting a batch size of 1024 with SimCSE triggers an "Out of Memory" (OOM) error during training, an issue our method (with a batch size of 256) does not encounter conversely.
>
> In light of these findings, we respectfully assert that **our methodology is not merely a "trick" analogous to employing baselines with larger batch sizes**.
>
> > *2. Very small improvement* and *mismatch performance for SimCSE-BERT model*
>
> > *2.1 Very small improvement*
>
> First, we *respectfully disagree* with the characterization that our performance improvements are very small.
>
> To provide a fair assessment, out of 12 different backbones:
> * Our model closely matched four backbones (with <0.2% performance difference).
> * HiCL surpassed another four backbones by a margin of 0.2% to 1%.
> * Notably, HiCL achieved a substantial improvement, exceeding 1%, on the remaining four backbones.
>
> The above results are verified on 3 different initializations and show good consistency. We mentioned between Line 900-907 that we do not re-tune the hyperparameters on different random seeds, which could explain why the improvement becomes smaller on 3-runs. We believe that by carefully fine-tuning the hyperparameters for each random seed, further enhancements can be achieved.
>
> We'd like to underscore that our method's primary value lies in its **seamless integration** with existing CL frameworks, irrespective of their augmentation methodologies or foundational architectures. This integration consistently drives improvements across two pivotal axes: encoding efficiency (shown in Section 3.3 and Appendix A.1) and overall performance.
>
> While we acknowledge that our model shows similar performance to the baselines on the four backbones, we urge reviewers to **adopt a holistic perspective when evaluating our contributions, not solely based on performance**. Performance metrics are just one facet of our contribution.  It’s worth noting that our proposed methodology also improves traditional CL’s encoding efficiency by a large margin.
>
> > *2.2 Mismatch Performance for SimCSE model*
>
> The discrepancy in performance for SimCSE models arises because we **unified the batch sizes to ensure a fair comparison** across baselines (using 256 for base models and 128 for large models).
>
> We reassessed the SimCSE-BERT_base model using its optimal hyperparameters (batch size 64) and the results are displayed below. **Regardless of whether we use SimCSE’s optimal settings or our uniform configuration, our method consistently outperforms the baseline.**
>
> |        |  batch  |  STS12 |   STS13    |STS14| STS15  | STS16 | STS-B |  SICK-R|   Avg. |
> |:-----:|:--------:|:----------:|:------------:|:--------:|:-----:|:--------:|:----------:|:------------:|:--------:|
> |Original SimCSE      |  64    | 68.40  | 82.41  | 74.38  | 80.91 | 78.56 | 76.85 | 72.23  | 76.25 |
> |Reproduced SimCSE| 256   | 66.92  | 78.44  | 71.15  | 79.50 | 77.71 | 75.50 | 68.76  | 74.00 |
> | + HiCL |                   256     | 69.04   | 80.68  |72.71   |80.39  |78.68| 76.96  |70.35    |75.54↑|
> | Reproduced SimCSE| 64   | 69.17   | 82.02  | 73.41  | 80.54 |78.36 |76.60  | 71.96   |76.01|
> |+HiCL | 64 | 69.44 | 82.10    | 74.48 | 81.62 | 78.77 | 77.75   | 70.26   | 76.35↑|
>
> As outlined in Lines 858-874, baselines vary in configurations like batch size. Recognizing that these factors significantly influence the final results, our aim was to assess different baselines under consistent conditions. To clarify, it would be misleading to evaluate, say, SimCSE-BERT_base with a batch size of 64 while assessing SNCSE-BERT_base with a batch size of 256. Such discrepancies could obscure the true reasons behind performance gaps.
>
> Lastly, we want to mention that some baselines benefitted from our standardized setup, displaying improved performance. For example, our reproduced SimCSE-RoBERTa_base jumped from an original 76.57 to 77.09. Likewise, SimCSE-RoBERTa_large saw an increase from 78.90 to 78.99.
>
> > *3. Generalizing to longer training corpus*
>
> We'd like to draw the reviewers' attention to Section 5.5, where we have already evaluated HiCL's performance on the training corpora with the longer sequence length.
>
> Specifically, we incorporated WikiText-103, which comprises 103 million tokens derived from 28,475 articles, with each article averaging 3617 tokens. Our findings indicate a more pronounced improvement for HiCL over the baseline when the training corpus is longer, with the enhancement increasing from +0.25% to +0.48%. We hope this answer could address your concern about train-test length mismatch setting (which is exactly train on long, test on short).
>
> We would greatly appreciate it if the reviewers could re-assess our work, especially if our explanations have clarified any previous ambiguities. We will incorporate further motivation in our revised manuscript.
>
> ---
> [1] Wu, Zhuofeng, Sinong Wang, Jiatao Gu, Madian Khabsa, Fei Sun, and Hao Ma. "Clear: Contrastive learning for sentence representation." arXiv preprint arXiv:2012.15466 (2020).
>
> [2] Wang, Hao, and Yong Dou. "SNCSE: contrastive learning for unsupervised sentence embedding with soft negative samples." In International Conference on Intelligent Computing, pp. 419-431. Singapore: Springer Nature Singapore, 2023.

---

### Official Review · Reviewer_69hi · 2023-08-10

**Soundness:** 3

**Excitement:**

4: Strong: This paper deepens the understanding of some phenomenon or lowers the barriers to an existing research direction.

**Paper Topic And Main Contributions:**

Main contributions: novel approaches to improve computational efficiency
- first encoding local segments and then aggregating to obtain global sequence representation
- this way, the authors claim to have improved training efficiency and also SOTA performance on seven STS tasks

**Reasons To Accept:**

- First to propose a hierarchical contrastive learning framework (which is a bold claim and surprising if true)
- Extensive ablation study and robustness checks
- SOTA performance (at least in some settings)

**Reasons To Reject:**

- Generally too strong claims ("First", "SOTA")
- Main results do not seem to strongly support the authors' claim that HiCL generally improves the baseline models
- Comparing just the average is a poor practice. I think the confidence intervals should be reported in order to judge whether improvement is statistically significant.


**Reproducibility:**

4: Could mostly reproduce the results, but there may be some variation because of sample variance or minor variations in their interpretation of the protocol or method.

**Reviewer Confidence:**

2: Willing to defend my evaluation, but it is fairly likely that I missed some details, didn't understand some central points, or can't be sure about the novelty of the work.

---

> ### Author Rebuttal · Authors · 2023-08-29
>
> We thank the reviewer for their encouraging feedback. We address the questions below:
>
> > *1. Too strong claims*
>
> Thanks for the suggestion! We will revise the corresponding claims in the next version. Specifically:
> * in Line 22 "achieving new state-of-the-art results" -> "gaining +0.2% on BERT_large and +0.44% on RoBERTa_large over prior best results"
> *  in Line 23 "we are the first to explore" -> "we explore"
>
> > *2. Main Results do not seem to strongly support the claim*
>
> First, we'd like to emphasize that our method's **primary value** lies in its **seamless integration** with existing CL frameworks, irrespective of their augmentation methodologies or foundational architectures. This integration consistently drives improvements across two pivotal axes: encoding efficiency and overall performance.
>
> Regarding efficiency enhancements, both theoretical analysis (Section 3.3) and practical tests (Appendix A.1) are provided to show our model’s general improvement over baselines.
>
> In terms of performance, let us recall the results in Table 1 on 12 backbones:
> * Our model (HiCL) closely aligns with four baselines, with a difference of less than 0.2%.
> * HiCL outpaces another four baselines, with gains between 0.2% to 1%.
> * Importantly, HiCL marks a notable uplift of over 1% for the last four baselines.
>
> These outcomes are consistent across three separate initializations. While we acknowledge that our model shows similar performance to the baselines on the four backbones, **we wish to remind that performance enhancement is just one facet of our contribution.** Significantly, our proposed approach enhances the encoding efficiency of traditional CL methods considerably.
>
> > *3. Whether the improvement is statistically significant*
>
> What we have done: we executed the models on three separate random seeds and presented the average performance alongside the standard deviation in Table 12. A more thorough analysis can be found in Lines 345-358 and Appendix A.4. Briefly, there's a notable consistency between the multi-run results and the single-run outcomes.
>
> We concur that the confidence interval provides a more meaningful measure of results' significance than the standard deviation. Acknowledging the importance of your input, we intend to delve deeper into the significance of improvements in the camera-ready version. We hope this could address the reviewer’s concern about whether the improvement is significant.

---

### Meta-Review · Area_Chair_ur54 · 2023-09-18

**Recommendation:** 3

**Metareview:**

This paper aims to improve unsupervised sentence representations and efficiency by proposing a hierarchical contrasting learning method. Reviewers praised the proposed methods, the thorough experiments, and clarity in the paper. Through detailed back and forth discussions some reviewer complaints were addressed and their scores raised. However, reviewers still found the method's motivation to be lacking and the performance gains to be slight.

---

### Decision · Program_Chairs · 2023-10-07

**Decision:**

Accept-Findings

**Comment:**

This paper aims to improve unsupervised sentence representations and efficiency by proposing a hierarchical contrasting learning method. Reviewers praised the proposed methods, the thorough experiments, and clarity in the paper. Through detailed back and forth discussions some reviewer complaints were addressed and their scores raised. However, reviewers still found the method's motivation to be lacking and the performance gains to be slight.